# Post Selection Inference with Incomplete Maximum Mean Discrepancy Estimator

**Makoto Yamada**[1,2,3,4]*, **Denny Wu**[5,6]*, **Yao-Hung Hubert Tsai**[7], **Hirofumi Ohta**[8],
**Ichiro Takeuchi**[9], **Ruslan Salakhutdinov**[7], **Kenji Fukumizu**[2,4]
Kyoto University[1], RIKEN AIP[2], JST PRESTO[3], Institute of Statistical Mathematics[4],
University of Toronto[5], Vector Institute[6], Carnegie Mellon University[7],
University of Tokyo[8], Nagoya Institute of Technology[9]
`myamada@i.kyoto-u.ac.jp,dennywu@cs.toronto.edu,yaohungt@cs.cmu.edu`
`ohtahirofumi@gmail.com, takeuchi.ichiro@nitech.ac.jp`
`rsalakhu@cs.cmu.edu, fukumizu@ism.ac.jp`

## Abstract

Measuring divergence between two distributions is essential in machine learning and statistics and has various applications including binary classification, change point detection, and two-sample test. Furthermore, in the era of big data, designing divergence measure that is interpretable and can handle high-dimensional and complex data becomes extremely important. In this paper, we propose a post selection inference (PSI) framework for divergence measure, which can select a set of statistically significant features that discriminate two distributions. Specifically, we employ an additive variant of maximum mean discrepancy (MMD) for features and introduce a general hypothesis test for PSI. A novel MMD estimator using the incomplete U-statistics, which has an asymptotically normal distribution (under mild assumptions) and gives high detection power in PSI, is also proposed and analyzed theoretically. Through synthetic and real-world feature selection experiments, we show that the proposed framework can successfully detect statistically significant features. Last, we propose a sample selection framework for analyzing different members in the Generative Adversarial Networks (GANs) family.

## 1 Introduction

Computing the divergence between two probability distributions is fundamental to machine learning and has many important applications such as binary classification (Friedman et al., 2001), change point detection (Yamada et al., 2013a; Liu et al., 2013), two-sample test (Gretton et al., 2012; Yamada et al., 2013b), and generative models such as generative adversarial networks (GANs) (Goodfellow et al., 2014; Li et al., 2015b; Nowozin et al., 2016), to name a few. Recently, interpreting the difference between distributions has become an important task in applied machine learning (Mueller & Jaakkola, 2015; Jitkrittum et al., 2016) since it can facilitate scientific discovery. For instance, in biomedical binary classification tasks, it is common to analyze which variables or features are different between two different distributions (classes).

The simplest approach to measure divergence between two probability densities would be parametric methods. For example, $t$-test can be used if the distributions to be compared are known and well-defined (Anderson, 2001). However, in many real-world problems, the property of distribution is not known *a priori*, and therefore model assumptions are likely to be violated. In contrast, non-parametric methods can be applied to any distributions without prior assumptions. The maximum mean discrepancy (MMD) (Gretton et al., 2012) is an example of non-parametric discrepancy measures and is defined as the difference between the mean embeddings of two distributions in a reproducing kernel Hilbert space (RKHS). For RKHSs associated with characteristic kernels (Sriperumbudur et al., 2011), the difference in mean embeddings defines a proper metric, and thus MMD can capture potentially non-linear difference between distributions (i.e., higher order moments).

---

*Equal contribution

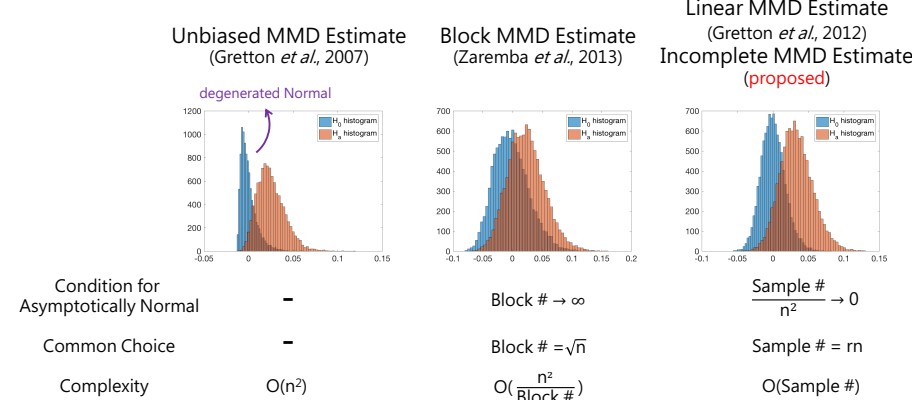

Figure 1: Comparison between Unbiased, Block, Linear, and Incomplete MMD estimation.

However, since MMD considers the entire $d$ dimensional vector, it is hard to interpret how individual features contribute to the discrepancy.

To deal with the interpretability issue, divergence measure with feature selection has been actively studied (Yamada et al., 2013a; Mueller & Jaakkola, 2015; Jitkrittum et al., 2016). For instance, in Mueller & Jaakkola (2015), the Wasserstein divergence is employed as a divergence measure and $\ell_1$ regularizer is used for feature selection. However, these approaches focus on detecting a set of features that discriminate two distributions. But for scientific discovery applications such as biomarker discovery, it might be preferable to test the significance of each selected feature (e.g., one biomarker). A naive approach would be to select features from one dataset and then test the selected features using the same data. However, in such case, *selection bias* is included and thus the false positive rate cannot be controlled. Therefore, it is crucial in hypothesis testing that the selection event should be taken into account to correct the bias. To the best of our knowledge, there is not yet an existing framework that tests the significance of selected features that distinguish between two different distributions on the same dataset.

In this paper, we propose `mmdInf`, a *post selection inference* (PSI) algorithm for distribution comparison, which finds a set of *statistically significant* features that discriminate between two distributions. `mmdInf` enjoys several compelling properties. First, it is a non-parametric method based on kernels, and thus it can detect features that distinguish various types of distributions. Second, the proposed framework is general and can be applied to not only feature selection but also other distribution comparison problems, such as dataset comparison. In `mmdInf`, we employ the recently developed PSI algorithm (Lee et al., 2016) and use MMD as a divergence measure. However, the standard empirical squared MMD estimator has a degenerated null distribution, which violates the requirement of a proper Normal distribution for the current PSI algorithm. To address this issue, we apply the block estimate (Zaremba et al., 2013) and the linear estimate (Gretton et al., 2012) of MMD. Furthermore, we propose a new empirical estimate of MMD based on the incomplete U-statistics (incomplete MMD estimator) (Blom, 1976; Janson, 1984; Lee, 1990) and show that it asymptotically follows the normal distribution, and has much greater detection power (compared to the existing estimators of MMD) in PSI. Finally, we propose a framework to analyze different members in the GAN (Goodfellow et al., 2014) family based on `mmdInf`. We elucidate the theoretical properties of the incomplete U-statistics estimate of MMD and show that `mmdInf` can successfully detect significant features through feature selection experiments.

The contributions of our paper are summarized as follows: **(1)** We propose a non-parametric PSI algorithm `mmdInf` for distribution comparison. **(2)** We propose the incomplete MMD estimator and investigate its theoretical properties (see Figure 1). **(3)** We propose a sample selection framework based on `mmdInf` that can be used for analyzing generative models.

## 2 RELATED WORK

Divergence measures in general can be divided into two categories: $f$-divergence (Ali & Silvey, 1966), such as the Kullback-Leibler divergence (Cover & Thomas, 2012) or the $\alpha$-divergence (Rényi et al., 1961; Póczos & Schneider, 2011), and integral probability metric (Müller, 1997), including

the total variation distance (Shorack & Shorack, 2000) and the Wasserstein distance in its dual (Villani, 2008). In this work we consider the maximum mean discrepancy (MMD) (Gretton et al., 2012), which falls into the second category. Although these divergence measures can be used for testing the discrepancy between $p(\boldsymbol{x})$ and $q(\boldsymbol{x})$, it is hard to test the significance of one of $k$ selected features from the entire $d$ features, where the setup is useful for scientific discovery tasks such as biomarker discovery. Recently, a MMD-based change point detection algorithm, which can compute the $p$-value of the largest score, has been proposed (Li et al., 2015a). However, this method can only test the largest score (i.e., one-feature). Thus, it is not clear whether the approach can be extended to feature selection in general.

A novel testing framework for the post selection inference (PSI) has been recently proposed (Lee et al., 2016), in which statistical inference after feature selection with Lasso (Tibshirani, 1996) is investigated. This work shows that statistical inference conditioned on the selection event can be done for linear regression models with Gaussian noise if the selection event can be written as a set of linear constraints. However, the PSI algorithm needs to assume a Gaussian response, which is a relatively strong assumption, and consequently it cannot be directly applied for non-Gaussian output problems such as classification. To deal with this issue, a kernel based PSI algorithm (`hsicInf`) for *independence test* has been proposed (Yamada et al., 2018), in which the Hilbert-Schmidt Independence Criterion (HSIC) (Gretton et al., 2005) is employed to measure the independence between input and its output, and thus significance test can be performed on non-Gaussian data. In this paper, we propose an alternative kernel based inference algorithm for *distribution comparison* called `mmdInf`, which can be used for both feature selection and binary classification.

In Sections 3.4 and 5.3, we manifest how we analyze different members in the GANs (Goodfellow et al., 2014) family. Here, we discuss several evaluation metrics that have been proposed to compare generative models. For example, the inception scores (Salimans et al., 2016) and the mode scores (Che et al., 2016) measure the quality and diversity of the generated samples, but they were not able to detect overfitting and mode dropping / collapsing for generated samples. The Frechet inception distance (FID) (Heusel et al., 2017) defines a score using the first two moments of the real and generated distributions, whereas the classifier two-sample tests (Lopez-Paz & Oquab, 2016) considers the classification accuracy of a binary classifier as a statistic for two-sample test. Although the above metrics are reasonable in terms of discriminability, robustness, and efficiency, the distances between samples are required to be computed in a suitable feature space. We can also use the kernel density estimation (KDE); or more recently, Wu et al. (2016) proposed to apply the annealed importance sampling (AIS) to estimate the likelihood of the decoder-based generative models. Nevertheless, these approaches need the access to the generative model for computing the likelihood, which are less favorable comparing to the model agnostic approaches which rely only on a finite generated sample set. On the other hand, the maximum mean discrepancy (MMD) (Gretton et al., 2012) is preferred against its competitors (Sutherland et al., 2016; Huang et al., 2018). Therefore, we propose the `mmdInf` based GANs analysis framework.

# 3 PROPOSED METHOD (MMDINF)

In this section, we introduce a PSI algorithm with MMD (Gretton et al., 2012).

## 3.1 POST SELECTION INFERENCE (PSI)

Suppose we are given independent and identically distributed (i.i.d.) samples $\boldsymbol{X} = \{\boldsymbol{x}_i\}_{i=1}^m \in \mathbb{R}^{d \times m}$ from a $d$-dimensional distribution $p$ and i.i.d. samples $\boldsymbol{Y} = \{\boldsymbol{y}_j\}_{j=1}^n \in \mathbb{R}^{d \times n}$ from another $d$-dimensional distribution $q$. Our goal is to find $k < d$ features that differentiate between $\boldsymbol{X}$ and $\boldsymbol{Y}$ and test whether each selected feature is *statistically significant*.

Let $\mathcal{S}$ be a set of selected features, we consider the following hypothesis test:

- $H_0$: $\sum_{s=1}^d \eta_s \widehat{D}(\boldsymbol{X}^{(s)}, \boldsymbol{Y}^{(s)}) = 0 \mid \mathcal{S}$ was selected,

- $H_1$: $\sum_{s=1}^d \eta_s \widehat{D}(\boldsymbol{X}^{(s)}, \boldsymbol{Y}^{(s)}) \neq 0 \mid \mathcal{S}$ was selected,

where $\widehat{D}(\boldsymbol{X}^{(s)}, \boldsymbol{Y}^{(s)})$ is the estimated discrepancy measure for the selected feature $s$ and $\boldsymbol{\eta} = [\eta_1, \ldots, \eta_d]^\top \in \mathbb{R}^d$ is an arbitrary pre-defined parameter. To test the $s$-th feature, we can set $\boldsymbol{\eta}$ as a unit vector whose $s$-th position is 1 and zero otherwise.

We employ the post-selection inference (PSI) framework (Lee et al., 2016) to test the hypothesis. Thanks to the cumulative distribution of Theorem 1, we can compute the $p$-value of $\sum_{s=1}^{d} \eta_s \widehat{D}(\boldsymbol{X}^{(s)}, \boldsymbol{Y}^{(s)})$ under feature selection.

**Theorem 1** *(Lee et al., 2016) Suppose that $\boldsymbol{z} \sim \mathcal{N}(\boldsymbol{\mu}, \boldsymbol{\Sigma})$, and the feature selection event can be expressed as $\boldsymbol{A}\boldsymbol{z} \leq \boldsymbol{b}$ for some matrix $\boldsymbol{A}$ and vector $\boldsymbol{b}$, then for any given feature represented by $\boldsymbol{\eta} \in \mathbb{R}^n$ we have*

$$F_{\boldsymbol{\eta}^\top \boldsymbol{\mu}, \boldsymbol{\eta}^\top \boldsymbol{\Sigma} \boldsymbol{\eta}}^{[V^-(\boldsymbol{A}, \boldsymbol{b}), V^+(\boldsymbol{A}, \boldsymbol{b})]}(\boldsymbol{\eta}^\top \boldsymbol{z}) \quad | \quad \boldsymbol{A}\boldsymbol{z} \leq \boldsymbol{b} \sim \text{Unif}(0, 1),$$

*where $F_{\mu, \sigma^2}^{[a,b]}(x)$ is the cumulative distribution function (CDF) of a truncated normal distribution with mean $\mu$ and variance $\sigma^2$ truncated at [a,b]. Given that $\boldsymbol{\alpha} = \boldsymbol{A} \frac{\boldsymbol{\Sigma} \boldsymbol{\eta}}{\boldsymbol{\eta}^\top \boldsymbol{\Sigma} \boldsymbol{\eta}}$, the lower and upper truncation points are given by*

$$V^-(\boldsymbol{A}, \boldsymbol{b}) = \max_{j: \boldsymbol{\alpha}_j < 0} \frac{b_j - (\boldsymbol{A}\boldsymbol{z})_j}{\boldsymbol{\alpha}_j} + \boldsymbol{\eta}^\top \boldsymbol{z}, \;\; V^+(\boldsymbol{A}, \boldsymbol{b}) = \min_{j: \boldsymbol{\alpha}_j > 0} \frac{b_j - (\boldsymbol{A}\boldsymbol{z})_j}{\boldsymbol{\alpha}_j} + \boldsymbol{\eta}^\top \boldsymbol{z}.$$

**Feature Selection with Discrepancy Measure:** Assume we have an estimate of a discrepancy measure for each feature: $\boldsymbol{z} = [\widehat{D}(\boldsymbol{X}^{(1)}, \boldsymbol{Y}^{(1)}), \ldots, \widehat{D}(\boldsymbol{X}^{(d)}, \boldsymbol{Y}^{(d)})]^\top \in \mathbb{R}^d \sim \mathcal{N}(\boldsymbol{\mu}, \boldsymbol{\Sigma})$, where $\widehat{D}(\cdot, \cdot)$ has large positive value when two distribution are different. We select the top-$k$ features with largest discrepancy scores. We denote the index set of the selected $k$ features by $\mathcal{S}$, and that of the unselected $\bar{k} = d - k$ features by $\bar{\mathcal{S}}$. This feature selection event can be characterized by

$$\widehat{D}(\boldsymbol{X}^{(s)}, \boldsymbol{Y}^{(s)}) \geq \widehat{D}(\boldsymbol{X}^{(\ell)}, \boldsymbol{Y}^{(\ell)}), \;\; \text{for all } (s, \ell) \in \mathcal{S} \times \bar{\mathcal{S}}.$$

Note that we have in total $k \cdot \bar{k}$ constraints.

The selection event can be rewritten as $\boldsymbol{A}_{s,\ell} \boldsymbol{z} \leq 0$, for all $(s, \ell) \in \mathcal{S} \times \bar{\mathcal{S}}$, where $\boldsymbol{A}_{s,\ell} = [0 \cdots 0 \underbrace{-1}_{s} 0 \cdots 0 \underbrace{1}_{\ell} 0 \cdots 0]$ and $\boldsymbol{A}_{s,\ell}^\top \in \mathbb{R}^d$ is a row vector of $\boldsymbol{A} \in \mathbb{R}^{(k \cdot \bar{k}) \times d}$. Under such construction, $\boldsymbol{A}\boldsymbol{z} \leq \boldsymbol{b}$ can be satisfied by setting $\boldsymbol{b} = \boldsymbol{0}$.

## 3.2 Maximum Mean Discrepancy (MMD)

We employ the maximum mean discrepancy (MMD) (Gretton et al., 2012) as divergence measure.

Let $\mathcal{F}$ be the unit ball in a reproducing kernel Hilbert space (RKHS) and $K(\boldsymbol{x}, \boldsymbol{x}')$ the corresponding positive definite kernel with $\mathbb{E}_{\boldsymbol{x}, \boldsymbol{x}' \sim p}[\sqrt{K(\boldsymbol{x}, \boldsymbol{x})}] < \infty, \mathbb{E}_{\boldsymbol{x}, \boldsymbol{x}' \sim q}[\sqrt{K(\boldsymbol{x}, \boldsymbol{x})}] < \infty$, the squared population MMD is defined as

$$\text{MMD}^2[\mathcal{F}, p, q] = \mathbb{E}_{\boldsymbol{x}, \boldsymbol{x}'}[K(\boldsymbol{x}, \boldsymbol{x}')] - 2\mathbb{E}_{\boldsymbol{x}, \boldsymbol{y}}[K(\boldsymbol{x}, \boldsymbol{y})] + \mathbb{E}_{\boldsymbol{y}, \boldsymbol{y}'}[K(\boldsymbol{y}, \boldsymbol{y}')],$$

where $\mathbf{E}_{\boldsymbol{x}, \boldsymbol{y}}$ denotes the expectation over independent random variables $\boldsymbol{x}$ with distribution $p$ and $\boldsymbol{y}$ with distribution $q$. It has been shown that if the kernel function $K(\boldsymbol{x}, \boldsymbol{x}')$ is characteristic, then $\text{MMD}[\mathcal{F}, \boldsymbol{X}, \boldsymbol{Y}] = 0$ if and only if $p = q$ (Gretton et al., 2012). In the following, we introduce two existing MMD estimators and then propose the incomplete U-statistics MMD estimator.

**(Complete) U-statistics estimator (Gretton et al., 2012):** We use the Gaussian kernel: $K(\boldsymbol{x}, \boldsymbol{x}') = \exp\left(-\frac{\|\boldsymbol{x} - \boldsymbol{x}'\|_2^2}{2\sigma_x^2}\right)$, where $\sigma_x > 0$ is the Gaussian width. When $m = n$, the *complete* U-statistics of MMD is defined as

$$\text{MMD}_u^2[\mathcal{F}, \boldsymbol{X}, \boldsymbol{Y}] = \frac{1}{m(m-1)} \sum_{i \neq j} h(\boldsymbol{u}_i, \boldsymbol{u}_j),$$

where $h(\boldsymbol{u}, \boldsymbol{u}') = K(\boldsymbol{x}, \boldsymbol{x}') + K(\boldsymbol{y}, \boldsymbol{y}') - K(\boldsymbol{x}, \boldsymbol{y}') - K(\boldsymbol{x}', \boldsymbol{y})$ is the U-statistics kernel for MMD and $\boldsymbol{u} = [\boldsymbol{x}^\top \boldsymbol{y}^\top]^\top \in \mathbb{R}^{2d}$. However, since the complete U-statistics estimator of MMD is degenerated under $p = q$ and does not follow normal distribution, this estimator cannot be used in PSI.

---

**Algorithm 1** `mmdInf` (Feature Selection) $H_0$ : $\sum_{s=1}^{d} \eta_s \text{MMD}_{inc}^2(\boldsymbol{X}^{(s)}, \boldsymbol{Y}^{(s)}) = 0 \,|\, \mathcal{S}$ was selected

---

**Input:** Data matrices $\boldsymbol{X} = [\boldsymbol{X}_{sel} \, \boldsymbol{X}_c] \in \mathbb{R}^{d \times m}$ and $\boldsymbol{Y} = [\boldsymbol{Y}_{sel} \, \boldsymbol{Y}_c] \in \mathbb{R}^{d \times n}$. Params: $\boldsymbol{\eta} \in \mathbb{R}^d$ and $k$.
1: Compute $\widehat{\boldsymbol{z}} = [\text{MMD}_{inc}^2(\boldsymbol{X}_{sel}^{(1)}, \boldsymbol{Y}_{sel}^{(1)}), \dots, \text{MMD}_{inc}^2(\boldsymbol{X}_{sel}^{(d)}, \boldsymbol{Y}_{sel}^{(d)})]^\top \in \mathbb{R}^d$.
2: Select $k$ features (i.e., $k = |\mathcal{S}|$) and compute $\boldsymbol{A}$ using $\widehat{\boldsymbol{z}}$ and $\widehat{\boldsymbol{\Sigma}}$ using $\boldsymbol{X}_c$ and $\boldsymbol{Y}_c$. Set $\boldsymbol{b} = \boldsymbol{0}$.
3: Compute $p$-value as $p = 1 - F_{0, \boldsymbol{\eta}^\top \widehat{\boldsymbol{\Sigma}} \boldsymbol{\eta}}^{[V^-(\boldsymbol{A}, \boldsymbol{b}), V^+(\boldsymbol{A}, \boldsymbol{b})]}(\boldsymbol{\eta}^\top \widehat{\boldsymbol{z}})$.

---

**Block estimator (Zaremba et al., 2013):** The block estimate of MMD is given by

$$\text{MMD}_b^2[\mathcal{F}, \boldsymbol{X}, \boldsymbol{Y}] = \frac{B_1}{n} \sum_{i=1}^{n/B_1} \text{MMD}_u^2[\mathcal{F}, \boldsymbol{X}_i, \boldsymbol{Y}_i],$$

where $\boldsymbol{X} = [\boldsymbol{X}_1, \dots, \boldsymbol{X}_{m/B_1}]$, $\boldsymbol{Y} = [\boldsymbol{Y}_1, \dots, \boldsymbol{Y}_{n/B_2}]$, and $\boldsymbol{X}_i \in \mathbb{R}^{d \times B_1}$ and $\boldsymbol{Y}_i \in \mathbb{R}^{d \times B_2}$ are $i$-th partitioned data. Here, we assume that the number of blocks $m/B_1 = n/B_2$ is an integer and $B = B_1 = B_2$. A commonly used heuristic for the block size is to set $B = \sqrt{n}$. This estimator asymptotically follows the normal distribution when $B_1$ and $B_2$ are finite and $m$ and $n$ go to infinity. The block estimator can be used for PSI, but the variance and normality depends on the partition of $\boldsymbol{X}$ and $\boldsymbol{Y}$. Specifically, when the total number of samples is small, then a small block size would result in high variance, whereas larger block size tends to result in non-Gaussian response.

**Incomplete U-statistics estimator:** The described problems of the block-estimator motivated us to design a new MMD estimator that is normally distributed and has smaller variance. We therefore propose an MMD estimator based on the incomplete U-statistics (Blom, 1976; Janson, 1984; Lee, 1990). The incomplete U-statistics estimator of MMD is given by

$$\text{MMD}_{inc}^2[\mathcal{F}, \boldsymbol{X}, \boldsymbol{Y}] = \frac{1}{\ell} \sum_{(i,j) \in \mathcal{D}} h(\boldsymbol{u}_i, \boldsymbol{u}_j),$$

where $\mathcal{D}$ is a subset of $\mathcal{S}_n = \{(i,j)\}_{i \neq j}$, and $\ell = |\mathcal{D}|$. $\mathcal{D}$ can be fixed design or random design. In particular, if we design $\mathcal{D}$ as $\mathcal{D} = \{(1,2),(3,4),\dots,(n-1,n)\}$ then the incomplete U-statistic corresponds to the linear-time MMD estimator (Gretton et al., 2012).

### 3.3 COVARIANCE MATRIX ESTIMATION

For PSI, we need to estimate the covariance matrix $\boldsymbol{\Sigma}$ from data.

**Block estimator** To estimate $\boldsymbol{\Sigma}$, we first compute $\boldsymbol{H}_b \in \mathbb{R}^{d \times (n/B)}$ whose elements are the mean-subtracted MMDs. Then, we regard $\boldsymbol{H}_b$ as the data matrix and use a standard covariance estimator for estimating $\widehat{\boldsymbol{\Sigma}}_b$. Note that for small $n$, the number of blocks $n/B$ tends to be smaller than the dimension $d$, and thus, the estimation accuracy of covariance matrix is low and affects the detection accuracy. To handle this issue, we employ the POET algorithm (Fan et al., 2013).

**Incomplete U-statistics estimator** To estimate $\boldsymbol{\Sigma}$, we first compute $\boldsymbol{H}_{inc} \in \mathbb{R}^{d \times \ell}$ whose elements are the mean-subtracted U-stat kernels. Then, we regard $\boldsymbol{H}_{inc}$ as the data matrix and can use a standard covariance estimator to compute $\widehat{\boldsymbol{\Sigma}}_{inc}$.

The estimation performance of the covariance estimation for the block estimate heavily depends on the block size $B$. That is, if $B$ is large, we need to estimate $\widehat{\boldsymbol{\Sigma}}_b$ from a small number of samples. On the other hand, in the incomplete U-statistics estimator, we can set $\ell = rn \gg n/B$, where $r > 0$ is a constant (See Section 4). Thus, in practice, the estimated covariance matrix tends to be more accurate than the block estimator counterpart. Figure 6 in the supplementary material shows the MSEs between the true and the estimated covariance matrices of the incomplete estimate and the block estimate, respectively. As can be seen, the error of the incomplete estimate is smaller than that of the block estimate, which is an advantage of the incomplete MMD over the block MMD in PSI.

### 3.4 ADDITIONAL APPLICATIONS (GANs ANALYSIS)

The proposed PSI framework is general and can be used for not only feature selection but also sample selection. In generative modeling, the generated distribution should match the data distribution;

in other words, the discrepancy between the generated data and real data should be small. We can therefore apply the selective inference algorithm and use the significance value to evaluate the generation quality. In this paper we apply `mmdInf` to compare the performance of GANs. We first select the model whose generated samples has the smallest MMD score with the real data and then perform the hypothesis test.

Let $\boldsymbol{x}_i^{(s)} \in \mathbb{R}^p$ be a feature vector generated by $s$-th GAN model with random seed $i$ and $\boldsymbol{y}_j \in \mathbb{R}^p$ is a feature vector of an original image. Image features can be extracted by pre-trained Resnet (He et al., 2016) or auto-encoders. The hypothesis test can be written as

- $H_0$: $\text{MMD}_{inc}^2[\mathcal{F}, \boldsymbol{X}^{(s)}, \boldsymbol{Y}] = 0 \mid s$ generates samples closest to the real distribution,
- $H_1$: $\text{MMD}_{inc}^2[\mathcal{F}, \boldsymbol{X}^{(s)}, \boldsymbol{Y}] \neq 0 \mid s$ generates samples closest to the real distribution.

Since we want to test the best generator that minimizes the discrepancy between generated and true samples (e.g., low MMD score), this sample selection event can be characterized by

$$\text{MMD}_{inc}^2[\mathcal{F}, \boldsymbol{X}^{(s)}, \boldsymbol{Y}] \leq \text{MMD}_{inc}^2[\mathcal{F}, \boldsymbol{X}^{(\ell)}, \boldsymbol{Y}], \quad (s, \ell) \in \mathcal{S} \times \bar{\mathcal{S}}.$$

## 4 THEORETICAL ANALYSIS OF INCOMPLETE MMD

We investigate the theoretical properties of the incomplete MMD estimator under the random design with replacement. For simplicity, we denote $\text{MMD}_{inc}^2[\mathcal{F}, \boldsymbol{X}, \boldsymbol{Y}] = \text{MMD}_{inc}^2$, $\text{MMD}^2[\mathcal{F}, p, q] = \text{MMD}^2$, $\text{MMD}_{inc}^2[\mathcal{F}, \boldsymbol{X}^{(s)}, \boldsymbol{Y}^{(s)}] = \text{MMD}_{inc}^{2,(s)}$, and $\text{MMD}^2[\mathcal{F}, p^{(s)}, q^{(s)}] = \text{MMD}^{2,(s)}$, respectively. We introduce the conditional expectations for the U-statistic kernel $h(\boldsymbol{u}, \boldsymbol{u}')$ as $h_1 = \mathbb{E}_{\boldsymbol{u}'}[h(\boldsymbol{u}, \boldsymbol{u}')]$ and $h_2 = h(\boldsymbol{u}, \boldsymbol{u}')$. See the supplementary material for proof.

**Theorem 2** *Let $c$ be the smallest integer such that $h_c \neq \text{MMD}^2$, and let $n$ and $\ell$ tend to infinity such that $\gamma = \lim_{n,\ell \to \infty} n^{-c}\ell$, $0 \leq \gamma \leq \infty$. For sampling with replacement, we have*

$$\begin{cases} \ell^{\frac{1}{2}}(\text{MMD}_{inc}^2 - \text{MMD}^2) \xrightarrow{D} \mathcal{N}(0, \sigma^2), & \text{if } \gamma = 0. \\ n^{\frac{c}{2}}(\text{MMD}_{inc}^2 - \text{MMD}^2) \xrightarrow{D} V, & \text{if } \gamma = \infty. \\ \ell^{\frac{1}{2}}(\text{MMD}_{inc}^2 - \text{MMD}^2) \xrightarrow{D} \gamma^{\frac{1}{2}}V + T, & \text{if } 0 < \gamma < \infty, \end{cases}$$

*where $V$ is the random variable of the limit distribution of $n^{\frac{c}{2}}(\text{MMD}_u^2 - \text{MMD}^2)$, $T$ is the random variable of $\mathcal{N}(0, \sigma^2)$, $\sigma^2 = Var(h(\boldsymbol{u}, \boldsymbol{u}'))$, and $T$ and $V$ are independent.*

**Corollary 3** *Assume $\lim_{n,\ell \to \infty} n^{-2}\ell = 0$ and $0 < \gamma = \lim_{n,\ell \to \infty} n^{-1}\ell < \infty$. For sampling with replacement, incomplete U-statistics estimator of MMD is asymptotically normally distributed as*

$$\begin{cases} \ell^{\frac{1}{2}}\text{MMD}_{inc}^2 \xrightarrow{D} \mathcal{N}(0, \sigma^2), & \text{if } p = q. \\ \ell^{\frac{1}{2}}(\text{MMD}_{inc}^2 - \text{MMD}^2) \xrightarrow{D} \mathcal{N}(0, \sigma^2 + \gamma\sigma_u^2), & \text{if } p \neq q. \end{cases}$$

*where $\sigma^2 = Var(h(\boldsymbol{u}, \boldsymbol{u}'))$ and $\sigma_u^2 = 4(\mathbb{E}_{\boldsymbol{u}}[(\mathbb{E}_{\boldsymbol{u}'}[h(\boldsymbol{u}, \boldsymbol{u}')]] - \mathbb{E}_{\boldsymbol{u},\boldsymbol{u}'}[h(\boldsymbol{u}, \boldsymbol{u}')])^2)$.*

**Corollary 4** *Assume $\lim_{n,\ell \to \infty} n^{-1}\ell = 0$. For sampling with replacement, the incomplete U-statistics estimator of MMD is asymptotically normally distributed as*

$$\ell^{\frac{1}{2}}(\text{MMD}_{inc}^2 - \text{MMD}^2) \xrightarrow{D} \mathcal{N}(0, \sigma^2).$$

Thus, in practice, by setting $\ell \ll n^2$, the incomplete estimator is asymptotically normal and therefore can be applied in PSI. More specifically, we can set $\ell = rn \ll n^2$, where $r$ is a small constant. In practice, we found that $r = 10$ works well in general.

Figure 2 shows the empirical distribution under $p = q$ and $p \neq q$ for the complete estimator, the block estimator and the incomplete estimator. As can be observed, the empirical distribution of the incomplete estimator is normal for small sampling parameter $r$, and becomes similar to its complete counterpart if $r$ is large; this is supported by Theorem 2 ($\gamma = \infty$).

Finally, the following theorem assure the joint normality of the $\boldsymbol{z}$ vector in Theorem 1.

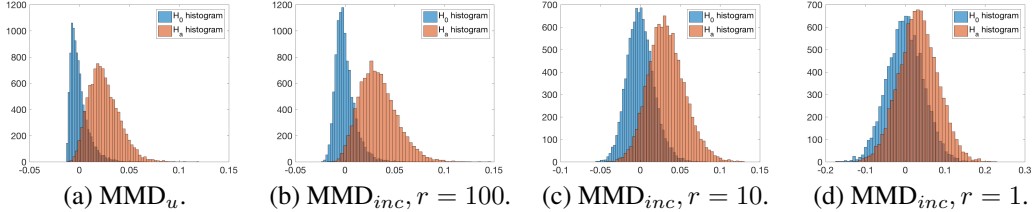

(a) $\text{MMD}_u$.     (b) $\text{MMD}_{inc}, r = 100$.     (c) $\text{MMD}_{inc}, r = 10$.     (d) $\text{MMD}_{inc}, r = 1$.

Figure 2: Empirical distribution under $p = q$ and $p \neq q$. (a) Complete U-statistics. (b)-(d): The incomplete MMD estimator with different sampling parameter $r$. For all plots, we fixed the number of samples as $n = 200$ and the dimensionality $d = 1$.

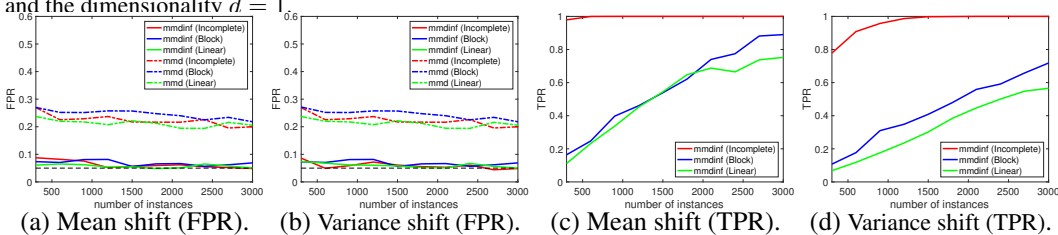

(a) Mean shift (FPR).    (b) Variance shift (FPR).    (c) Mean shift (TPR).    (d) Variance shift (TPR).

Figure 3: (a)(b): False positive rates at significant level $\alpha = 0.05$ of the proposed incomplete estimator, block estimator and linear estimator with/without PSI. For incomplete MMD, we set $\ell = 10n$. For block MMD, we set the block size $B = \sqrt{n}$. The MMD without PSI computes the $p$-values without adjusting for the selection. (c)(d): True positive rate comparison of the following three empirical estimates for `mmdInf`.

**Theorem 5** *Suppose $0 \leq \gamma < \infty$. Then,*

$$\ell^{1/2} \left\{ \left[ \text{MMD}_{inc}^{2,(1)}, \ldots, \text{MMD}_{inc}^{2,(d)} \right] - \left[ \text{MMD}^{2,(1)}, \ldots, \text{MMD}^{2,(d)} \right] \right\}$$

*converges to multivariate normal distribution.*

## 5 EXPERIMENTS

We compared `mmdInf` with a naive testing baseline (`mmd`), which first selects features using MMD and estimates corresponding $p$-values with the same data of feature selection without adjustment for the selection event. For `mmdInf`, we used the three MMD estimators: the linear-time MMD (Gretton et al., 2012), the block MMD (Zaremba et al., 2013), and the incomplete MMD. We used $1/2$ of data to calculate the covariance matrix of MMD and the rest to perform feature selection and inference. We fixed the number of selected features (prior to PSI) $k$ to 30. In PSI the significance of each of the 30 selected features (from ranking MMD) is computed and features with $p$-value lower than the significance level $\alpha = 0.05$ are selected as *statistically significant features*.

For block MMD, in each experiment we set the candidate of block size as $B = \{10, 20, 50\}$ . For incomplete MMD, in each experiment the ratio between number of pairs $(i, j)$ sampled to compute incomplete MMD score and sample size is fixed at $r = \frac{\ell}{n} \in \{0.5, 5, 10\}$. We reported the true positive rate (TPR) $\frac{k'}{k^*}$ where $k'$ is the number of true features selected by `mmdInf` or `mmd` and $k^*$ is the total number of true features in synthetic data. We further computed the false positive rate (FPR) $\frac{k''}{k - k^*}$ where $k''$ is the number of *non-true* features reported as positives. We ran all experiments over 200 times, and reported the average TPR and FPR.

### 5.1 SYNTHETIC EXPERIMENTS (PSI)

The number of features $d$ is fixed to 50, and for each feature, data is randomly generated following a Gaussian distribution with set mean and variance. 10 out of the 50 features are set to be significantly different by shifting the distribution of one class away from the other (mean or variance). More specifically, we generate the synthetic data as

**(a) Mean shift** $x \sim \mathcal{N}(\mathbf{0}_{50}, \mathbf{I}_{50})$, $y \sim \mathcal{N}(\boldsymbol{\mu}, \mathbf{I})$, $\boldsymbol{\mu} = [\mathbf{1}_{10}^\top \, \mathbf{0}_{40}^\top]^\top \in \mathbb{R}^{50}$,
**(b) Variance shift** $x \sim \mathcal{N}(\mathbf{0}_{50}, \mathbf{I}_{50})$, $y \sim \mathcal{N}(\mathbf{0}, \boldsymbol{\Sigma})$, $\boldsymbol{\Sigma} = \text{diag}([1.5\mathbf{1}_{10}^\top \, \mathbf{1}_{40}^\top]^\top)$,

where $\mathcal{N}(\boldsymbol{\mu}, \boldsymbol{\Sigma})$ is a multivariate normal distribution with mean $\boldsymbol{\mu} \in \mathbb{R}^d$ and covariance $\boldsymbol{\Sigma} \in \mathbb{R}^{d \times d}$, $\mathbf{1}_p \in \mathbb{R}^d$ is a vector whose elements are all one, $\mathbf{0}_d \in \mathbb{R}^d$ is a vector whose elements are all zero, and $\text{diag}(\boldsymbol{a}) \in \mathbb{R}^{d \times d}$ is a diagonal matrix whose diagonal elements are $\boldsymbol{a} \in \mathbb{R}^d$.

Table 1: Post selection inference experimental results for real-world datasets. The average TPR and FPR over 200 trials are reported.

| Datasets | $d$ | $n$ | Linear-Time | | Block | | | | | | Incomplete | | | | | |
| | | | | | $B = 10$ | | $B = 20$ | | $B = 50$ | | $r = 0.5$ | | $r = 5$ | | $r = 10$ | |
| | | | TPR | FPR | TPR | FPR | TPR | FPR | TPR | FPR | TPR | FPR | TPR | FPR | TPR | FPR |
| Diabetis | 8 | 768 | 0.05 | 0.06 | 0.20 | 0.04 | 0.40 | 0.12 | 0.46 | 0.18 | 0.13 | 0.06 | 0.52 | 0.07 | 0.65 | 0.10 |
| Wine (White) | 11 | 4898 | 0.13 | 0.06 | 0.26 | 0.01 | 0.56 | 0.04 | 0.68 | 0.09 | 0.32 | 0.06 | 0.71 | 0.06 | 0.79 | 0.06 |
| Wine (Red) | 11 | 1599 | 0.08 | 0.06 | 0.26 | 0.04 | 0.37 | 0.09 | 0.49 | 0.15 | 0.15 | 0.06 | 0.51 | 0.07 | 0.61 | 0.07 |
| Australia | 7 | 690 | 0.08 | 0.06 | 0.36 | 0.06 | 0.47 | 0.11 | 0.65 | 0.21 | 0.16 | 0.05 | 0.66 | 0.07 | 0.79 | 0.09 |

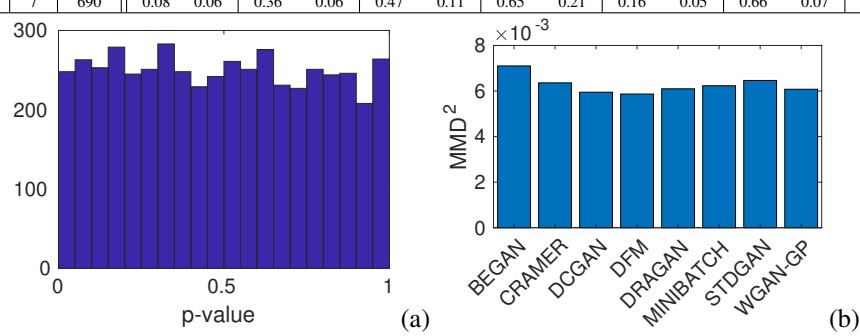

Figure 4: (a) Histogram of $p$-values over 5000 runs. (b) Averaged incomplete MMD scores.

Figure 3(a) and (b) show the FPRs of linear MMD, block MMD and incomplete MMD with or without PSI. As can be clearly seen, PSI successfully controls FPR with significance level $\alpha = 0.05$ for all the three estimators, whereas the naive approach tends to have higher FPRs. Figures 3 shows the TPRs of the synthetic data. In both cases, the TPR of incomplete MMD converges to 1 significantly faster than the the other two empirical estimates.

## 5.2 REAL-WORLD DATA (BENCHMARK)

We compared the proposed algorithm by using real-world datasets. Since it is difficult to decide what is a "true feature" in real-world data, we choose a few datasets for binary classification with small amount of features, and regard all the original features as *true*. We then concatenated random features to the *true* features (the total number of features $d = 100$). Table 1 shows TPRs and FPRs of `mmdInf` with different MMD estimators. It can be observed that the incomplete estimator significantly outperforms the other empirical estimates. Note that a higher TPR can be achieved with higher $r$, while the FPR is still controlled at 0.05 with $r = 5$. Additional comparison of different empirical estimates on real dataset (with varying sample size) can be found in Figures 8.

## 5.3 GANS ANALYSIS (SAMPLE SELECTION)

We also applied `mmdInf` for evaluating the generation quality of GANs. We trained BEGAN (Berthelot et al., 2017), DCGAN (Radford et al., 2015), STDGAN (Miyato et al., 2017), Cramer GAN (Bellemare et al., 2017), DFM (Warde-Farley & Bengio, 2016), DRAGAN (Kodali et al., 2017), Minibatch Discrimination GAN (Salimans et al., 2016), and WGAN-GP (Gulrajani et al., 2017), generated 5000 images (using Chainer GAN package [1] with CIFAR10 datasets), and extracted 512 dimensional features by pre-trained Resnet18 (He et al., 2016). For the true image sets, we subsampled 5000 images from CIFAR10 datasets and computed the 512 dimensional features using the same Resnet18. We then tested the difference between the generated images and the real images using `mmdInf` on extracted features (see Sec. 3.4).

We found that for all the members in the GAN family, the null hypothesis was rejected, i.e., the generated distribution and the real distribution are different. This result is consistent with the findings in (Sutherland et al., 2016), which demonstrate that optimized MMD has perfect discriminative power in GANs evaluation. As sanity check, we evaluated `mmdInf` by constructing an "oracle" generative model that generates real images from CIFAR10. Next, we randomly selected 5000 images (a disjoint set from the oracle generative images) from CIFAR10 in each trial, and set the sampling ratio to $r = 5$. Figure 4(a) shows the distribution of $p$-values computed by our algorithm. We can see that the $p$-values are distributed uniformly in the tests for the "oracle" generative model, which matches the theoretical result in Theorem 1. Thus the algorithm is able to detect the distribution difference and control the false positive rate. In other words, if the generated samples do not follow the original distribution, we can safely reject the null hypothesis with a given significance level $\alpha$.

---

[1] https://github.com/pfnet-research/chainer-gan-lib

Figure 4(b) shows the estimated MMD scores of each model. Based on the results, we could tell that DFM was the best model and DCGAN was the second best model to approximate the true distribution. However, the difference between various members is not obvious. Developing a validation pipeline based on `mmdInf` for GANs analysis would be one interesting line of future work.

## 6 CONCLUSION

In this paper, we proposed a novel statistical testing framework `mmdInf`, which can find a set of *statistically significant* features that can discriminate two distributions. Through synthetic and real-world experiments, we demonstrated that `mmdInf` can successfully find important features and/or datasets. We also proposed a method for sample selection based on `mmdInf` and applied it in the evaluation of generative models.

## 7    ACKNOWLEDGEMENT

We would like to thank Dr. Kazuki Yoshizoe, Prof. Yuta Umezu , and Prof. Suriya Gunasekar for fruitful discussions and valuable suggestions. MY was supported by the JST PRESTO program JPMJPR165A and partly supported by MEXT KAKENHI 16K16114. IT was partially supported by MEXT KAKENHI (17H00758, 16H06538), JST CREST (JPMJCR1302, JPMJCR1502), RIKEN Center for Advanced Intelligence Project, and JST support program for starting up innovation-hub on materials research by information integration initiative. KF was supported partly by JSPS KAK-ENHI 18K19793. YHHT and RS were supported in part by the DARPA grants D17AP00001 and FA875018C0150.

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

## Supplementary materials

### Theoretical Analysis of Incomplete MMD

We investigate the theoretical properties of the incomplete MMD estimator under the random design with replacement. For simplicity, we denote $\text{MMD}^2_{inc}[\mathcal{F}, \boldsymbol{X}, \boldsymbol{Y}] = \text{MMD}^2_{inc}$ and $\text{MMD}^2[\mathcal{F}, p, q] = \text{MMD}^2$, respectively.

We introduce the conditional expectations for the U-statistic kernel $h(\boldsymbol{u}, \boldsymbol{u}') = K(\boldsymbol{x}, \boldsymbol{x}') + K(\boldsymbol{y}, \boldsymbol{y}') - K(\boldsymbol{x}, \boldsymbol{y}') - K(\boldsymbol{x}', \boldsymbol{y})$:

$$h_1 = \mathbb{E}_{\boldsymbol{u}'}[h(\boldsymbol{u}, \boldsymbol{u}')], \ \ h_2 = h(\boldsymbol{u}, \boldsymbol{u}').$$

Then, we define $c$ be the smallest integer such that $h_c \neq \text{MMD}^2$.

For $p = q$, since the U-statistics is degenerated:

$$\begin{aligned}
h_1 &= \mathbb{E}_{\boldsymbol{u}'}[h(\boldsymbol{u}, \boldsymbol{u}')] \\
&= \mathbb{E}_{\boldsymbol{x}'}[K(\boldsymbol{x}, \boldsymbol{x}')] + \mathbb{E}_{\boldsymbol{y}'}[K(\boldsymbol{y}, \boldsymbol{y}')] - \mathbb{E}_{\boldsymbol{y}'}[K(\boldsymbol{x}, \boldsymbol{y}')] - \mathbb{E}_{\boldsymbol{x}'}[K(\boldsymbol{x}', \boldsymbol{y})] \\
&= 0,
\end{aligned}$$

and $\text{MMD}^2 = 0$, we have $c = 2$. On the other hand, since $h_1 \neq \text{MMD}^2$ for $p \neq q$, we have $c = 1$.

**Theorem 2** *Let $c$ be the smallest integer such that $h_c \neq \text{MMD}^2$, and let $n$ and $\ell$ tend to infinity such that $\gamma = \lim_{n, \ell \to \infty} n^{-c}\ell$, $0 \leq \gamma \leq \infty$. For sampling with replacement, we have*

$$\begin{cases}
\ell^{\frac{1}{2}}(\text{MMD}^2_{inc} - \text{MMD}^2) \xrightarrow{D} \mathcal{N}(0, \sigma^2), & \text{if } \gamma = 0. \\
n^{\frac{c}{2}}(\text{MMD}^2_{inc} - \text{MMD}^2) \xrightarrow{D} V, & \text{if } \gamma = \infty. \\
\ell^{\frac{1}{2}}(\text{MMD}^2_{inc} - \text{MMD}^2) \xrightarrow{D} \gamma^{\frac{1}{2}}V + T, & \text{if } 0 < \gamma < \infty.
\end{cases}$$

*where $V$ is the random variable of the limit distribution of $n^{\frac{c}{2}}(\text{MMD}^2_u - \text{MMD}^2)$, $T$ is the random variable of $\mathcal{N}(0, \sigma^2)$, $\sigma^2 = Var(h(\boldsymbol{u}, \boldsymbol{u}'))$, and $T$ and $V$ are independent.*

**Proof:** *Use Corollary 1 of Janson (1984) (or Theorem 1 of Lee (1990), pp. 200). In MMD, $c = 2$ for $p = q$ and $c = 1$ for $p \neq q$.* □

**Corollary 3** *Assume $\lim_{n, \ell \to \infty} n^{-2}\ell = 0$ and $0 < \gamma = \lim_{n, \ell \to \infty} n^{-1}\ell < \infty$. For sampling with replacement, the incomplete U-statistics estimator of MMD is asymptotically normally distributed as*

$$\begin{cases}
\ell^{\frac{1}{2}}\text{MMD}^2_{inc} \xrightarrow{D} \mathcal{N}(0, \sigma^2), & \text{if } p = q. \\
\ell^{\frac{1}{2}}(\text{MMD}^2_{inc} - \text{MMD}^2) \xrightarrow{D} \mathcal{N}(0, \sigma^2 + \gamma\sigma_u^2), & \text{if } p \neq q.
\end{cases}$$

*where $\sigma^2 = Var(h(\boldsymbol{u}, \boldsymbol{u}'))$ and $\sigma_u^2 = 4(\mathbb{E}_{\boldsymbol{u}}[(\mathbb{E}_{\boldsymbol{u}'}[h(\boldsymbol{u}, \boldsymbol{u}')]] - \mathbb{E}_{\boldsymbol{u}, \boldsymbol{u}'}[h(\boldsymbol{u}, \boldsymbol{u}')])^2)$.*

**Proof:** *Under $p = q$ ($c = 2$), since $\lim_{n, \ell \to \infty} n^{-2}\ell = 0$ and $\text{MMD}^2 = 0$, we can immediately obtain the limit distribution by Theorem 2. Under $p \neq q$ ($c = 1$), $\text{MMD}_u$ converges in distribution to a Gaussian according to (Gretton et al., 2007)*

$$n^{\frac{1}{2}}(\text{MMD}^2_u - \text{MMD}^2) \xrightarrow{D} \mathcal{N}(0, \sigma_u^2)$$

*where $\sigma_u^2 = 4(\mathbb{E}_{\boldsymbol{u}}[(\mathbb{E}_{\boldsymbol{u}'}[h(\boldsymbol{u}, \boldsymbol{u}')]] - \mathbb{E}_{\boldsymbol{u}, \boldsymbol{u}'}[h(\boldsymbol{u}, \boldsymbol{u}')])^2)$. Based on Theorem 2, under the given assumption, we can obtain the distribution of $MMD^2_{inc}$ since $T$ and $V$ are independent.* □

**Corollary 4** *Assume $\lim_{n, \ell \to \infty} n^{-1}\ell = 0$. For sampling with replacement, the incomplete U-statistics estimator of MMD is asymptotically normally distributed as*

$$\ell^{\frac{1}{2}}(\text{MMD}^2_{inc} - \text{MMD}^2) \xrightarrow{D} \mathcal{N}(0, \sigma^2).$$

**Proof:** *Since $\lim_{n, \ell \to \infty} n^{-1}\ell = 0$ and $\lim_{n, \ell \to \infty} n^{-2}\ell = 0$, the limit distribution of $\ell^{1/2}(\text{MMD}^2_{inc} - \text{MMD}^2)$ is $\mathcal{N}(0, \sigma^2)$ based on Theorem 2.* □

Thus, in practice, by setting $\ell \ll n^2$, the incomplete estimator is asymptotically normal and therefore can be applied in PSI. More specifically, we can set $\ell = rn \ll n^2$, where $r$ is a small constant. In practice, we found that $r = 10$ works well in general.

**Theorem 5** *Suppose $0 \leq \gamma < \infty$. Then,*

$$\ell^{1/2} \left\{ \left[ \mathrm{MMD}_{inc}^{2,(1)}, \ldots, \mathrm{MMD}_{inc}^{2,(d)} \right] - \left[ \mathrm{MMD}^{2,(1)}, \ldots, \mathrm{MMD}^{2,(d)} \right] \right\}$$

*converges to multivariate normal distribution.*

*Proof: We use the fact that convergence in distribution of multivariate random variables results in convergence in distribution of univariate random variables. From Theorem29.4 in Billingsley (2008)(Cramér-Wold device), it is sufficient to prove that for any $\boldsymbol{\eta} = [\eta_1, \ldots, \eta_d]^\top \in \mathbb{R}^d$, $\sum_{s=1}^d \eta_s \mathrm{MMD}_{inc}^2[\mathcal{F}, \boldsymbol{X}^{(s)}, \boldsymbol{Y}^{(s)}] \xrightarrow{D} \sum_{s=1}^d \eta_s Z_s$, where $Z_s, s = 1, 2, \ldots, d$ are normal distributions. Since each incomplete U-statistic $\mathrm{MMD}_{inc}^2[\mathcal{F}, \boldsymbol{X}^{(s)}, \boldsymbol{Y}^{(s)}]$ converges to normal distribution derived in Corollary 3 when $0 \leq \gamma < \infty$, and from the continuous mapping theorem for $g(\boldsymbol{x}) = \boldsymbol{\eta}^\top \boldsymbol{x}$, we obtain the desired result.*

ILLUSTRATIVE EXPERIMENTS

Figure 5 shows the empirical distribution under $p = q$ and $p \neq q$ for the complete estimator, the block estimator and the incomplete estimator. As can be observed, the empirical distribution of the incomplete estimator is normal for small sampling parameter $r$, and becomes similar to its complete counterpart if $r$ is large; this is supported by Theorem 2 ($\gamma = \infty$). Moreover, compared to the block estimator, the incomplete estimator tends to have a better trade-off between variance and normality.

Figure 6 shows the MSEs between the true and the estimated covariance matrices of the incomplete estimate and the block estimate, respectively. As can be seen, the error of the incomplete estimate is smaller than that of the block estimate, which is an advantage of the incomplete MMD over the block MMD in PSI. Moreover, for the incomplete U-statistics estimator, we have $\ell = rn$ samples to estimate the covariance matrix, while we have only $n/B$ samples for the block estimate. That is, the number of samples that can be used for covariance estimation of the incomplete U-statistics estimator is $rB$ times larger than that of the block estimator. For example, for $n = 1000$, we can use 5000 samples for the incomplete U-statistics, while we have only 50 samples for the block estimate. This is problematic for high-dimensional case (e.g., $d = 100$), and the estimated covariance matrix is not full rank. To alleviate this issue, we employ the POET algorithm for the block estimator.

Figure 7(a) shows the Type II error comparison for two-sample test with one dimensional Gaussian mean shift data. The Type II error is computed when the Type I error is fixed at 0.05, and the incomplete MMD outperforms other estimators. Figure 7 (b) compares the computational time of the empirical estimates, and for small $r$ the computational time of incomplete MMD is much less than that of the block MMD. Overall, the incomplete MMD has favorable properties in practice.

Figure 8 show the true positive rate (TPR) and the false positive rate (FPR) of the wine (white) dataset experiment. Here, we change the number of samples as $200, 400, \ldots, 4000$. As can be seen, both PSI algorithms can control the FPR with 2000 samples. Moreover, it is clear that the incomplete estimator can get better performance than the block estimator in both TPR and FPR with smaller number of samples.

EFFECTIVENESS OF PSI

Here, we compared the PSI and non-PSI counterpart on benchmark data. As clearly seen, if we do not use the PSI, we cannot control FPR, while the proposed algorithm can successfully control FPR values.

| Datasets | $d$ | $n$ | Linear-Time | | Incomplete (without PSI) | | | | | | Incomplete (with PSI) | | | | | |
|---|---|---|---|---|---|---|---|---|---|---|---|---|---|---|---|---|
| | | | | | $r=0.5$ | | $r=5$ | | $r=10$ | | $r=0.5$ | | $r=5$ | | $r=10$ | |
| | | | TPR | FPR | TPR | FPR | TPR | FPR | TPR | FPR | TPR | FPR | TPR | FPR | TPR | FPR |
| Diabetis | 8 | 768 | 0.05 | 0.05 | 0.41 | 0.21 | 0.74 | 0.27 | 0.83 | 0.31 | 0.13 | 0.06 | 0.52 | 0.07 | 0.65 | 0.10 |
| Wine (White) | 11 | 4898 | 0.13 | 0.06 | 0.58 | 0.23 | 0.81 | 0.25 | 0.86 | 0.26 | 0.32 | 0.06 | 0.71 | 0.06 | 0.79 | 0.06 |
| Wine (Red) | 11 | 1599 | 0.08 | 0.06 | 0.39 | 0.24 | 0.65 | 0.26 | 0.71 | 0.29 | 0.15 | 0.06 | 0.51 | 0.07 | 0.61 | 0.07 |
| Australia | 7 | 690 | 0.08 | 0.06 | 0.52 | 0.21 | 0.85 | 0.25 | 0.91 | 0.30 | 0.16 | 0.05 | 0.66 | 0.07 | 0.79 | 0.09 |

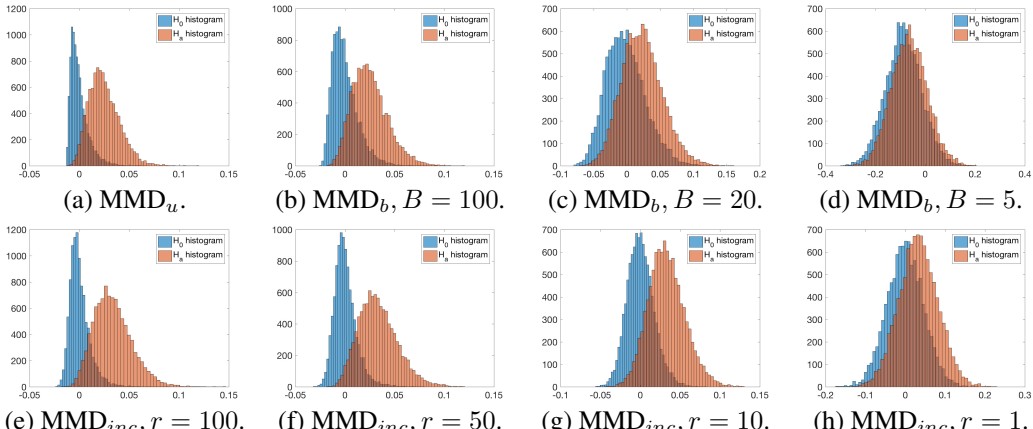

Figure 5: Empirical distribution under $p = q$ and $p \neq q$. (a) Complete U-statistics. (b)-(d): The block MMD estimator with different block parameter $B$. (e)-(h): The incomplete MMD estimator with different sampling parameter $r$. For all plots, we fixed the number of samples as $n = 200$ and the dimensionality $d = 1$.

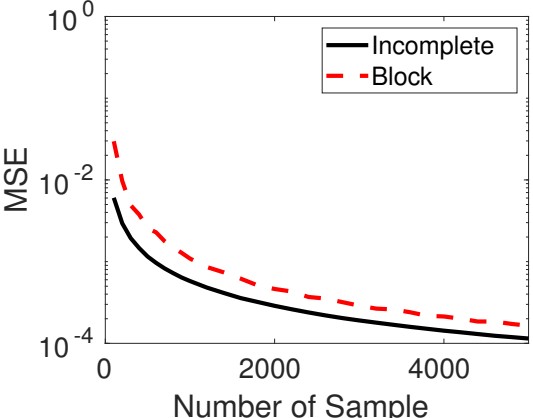

Figure 6: Covariance estimation error with respect to the number of samples. We fixed $B = 20$ and $\ell = 5n$.

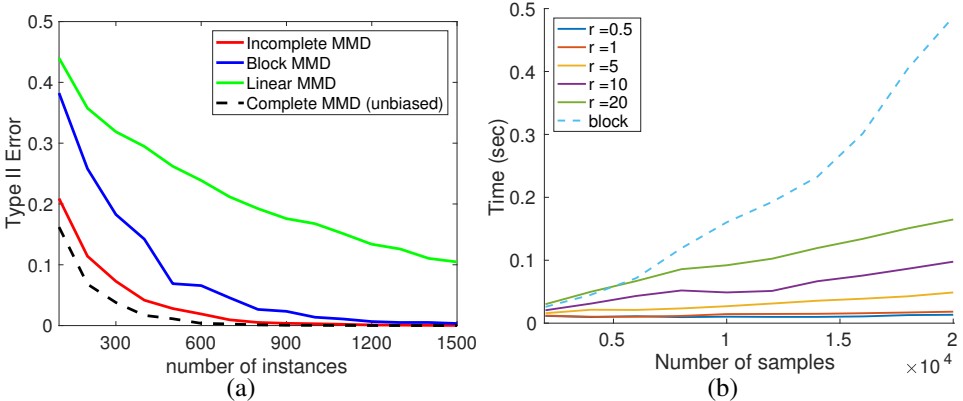

Figure 7: (a): Type II error comparison. We change the sample size $n = [100, 200 \ldots, 1500]$ and compute the type II error of the four empirical estimates of MMD when the type I error is controlled at 0.05. For incomplete MMD, we use $r = 10$. For the block MMD, we use $B = \sqrt{n}$. (b): Computational time comparison. We change the sample size $n = [2000, 4000, \ldots, 20000]$ and compute the incomplete MMD and the block MMD, respectively. For incomplete MMD, we use $r = [0.5, 1, 5, 10, 20]$. For the block MMD, we use $B = \sqrt{n}$. Incomplete MMD with $r = 0.5$ (i.e., $\ell = n/2$) can be regarded as the linear-time MMD estimator (Gretton et al., 2012).

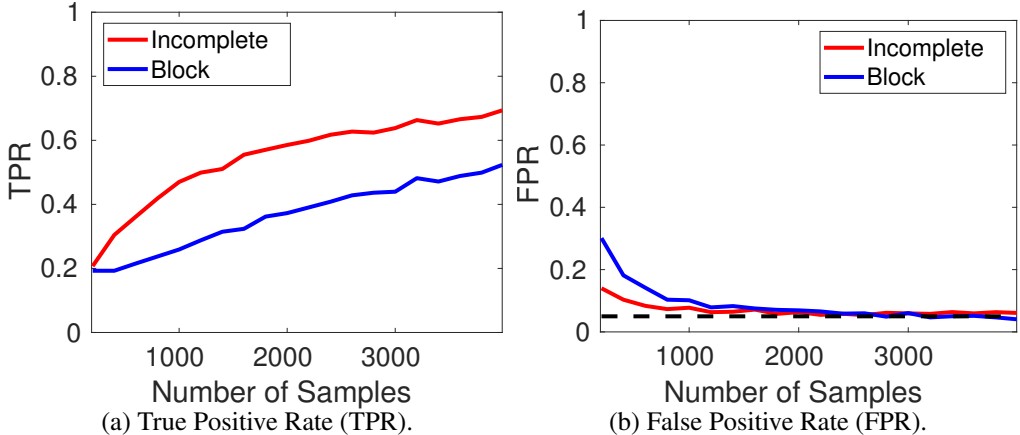

Figure 8: Wine (White) dataset experiment with varying number of samples. (a): True Positive Rate (TPR). (b): False Positive Rate (FPR). In this experiment, we set $\ell = 5n$ and $B = 20$, respectively.

