# OpenReview forum: "Post Selection Inference with Incomplete Maximum Mean Discrepancy Estimator"
_ICLR.cc/2019/Conference_

### Official Review · AnonReviewer3 · 2018-11-02
**clearly written, nice work**

**Rating:** 8
**Confidence:** 4

**Review:**

The authors focus on the selection problem of k statistically significant features discriminating 2 probability distributions accessible via samples. They propose a non-parametric approach under the PSI (post selection inference) umbrella using MMD (maximum mean discrepancy) as a discrepancy measure between probability distributions. The idea is to apply (asymptotically) normal MMD estimators, rephrase the top-k selection problem as a linear constraint, and reduce the problem to Lee et al., 2016. The efficiency of the approach is illustrated on toy examples and in GAN (generative adversarial network) context. The technique complements the PSI-based independence testing approach recently proposed by Yamada et al., 2018.

The submission is a well-organized, clearly written, nice contribution; it can be relevant to the machine learning community.

Below I enlist a few suggestions to improve the manuscript:
-Section 1: The notion of characteristic kernel (kernel when MMD is metric) has not been defined, but it was referred to. 'Due to the mean embeddings in RKHS, all moment information is stored.': This sentence is somewhat vague.
-Section 1: 'MMD can be computed in closed form'. This is rarely the case (except for e.g. Gaussian distributions with Gaussian or polynomial kernels). I assume that the authors wanted refer to the estimation of MMD.
-Section 1: 'K nearest neighbor approaches (Poczos & Schneider, 2011)'. The citation to this specific estimator can go under alpha-divergences. The Wasserstein metric could also be mentioned.
-Section 3.1: k is used to denote the number of selected features and also the kernel used in MMD. I suggest using different notations.
-Theorem 1: '\Phi is the CDF...'. There is no \Phi in the theorem.
-Section 3.2: The existence of MMD (mean embedding) requires certain assumptions: E_{x\sim p}\sqrt{k(x,x)} < \infty, E_{x\sim q}\sqrt{k(x,x)} < \infty.
-Section 3.2.: block estimator: 'B_1 and B_2 are finite'. 'fixed'?
-Section 3.2.: MMD_{inc}:
   i) 'S_{n,k}': k looks superfluous.
   ii) 'l': it has not been introduced (cardinality of D).
-Section 3.3: typo: 'covraiance' (2x)
-Section 3.3: Fan et al. 2013: The citation can go to \citep{}.
-Theorem 2:
   i)'c' is left undefined.
   ii)Comma is missing before 'where'.
   iii)\xrightarrow{d} (Theorem 2, Corollary 3-4): Given that 'd' also denotes dimension in the submission, I suggest using a different notation for convergence in distribution.
-At the introduction of block-MMD the block size (B) was fixed, while in the experiments (e.g. Figure 3) it is growing with the sample size (B=\sqrt{n}). The assumption on B should be clearly stated.
-Section 5.1: (b) mean shift: comma is missing before 'where'.
-References:
   i) Abbreviations and names in the titles should be capitalized (such as cramer, wasserstein, hilbert-schmidt, gan, nash).
   ii) Scholkopf should be Sch\{"o}lkopf (in the ALT 2005 work).
   iii) 'Exact post-selection inference, with application to the lasso': All the authors are listed; 'et al.' is not needed.

---

> ### Author Response · Authors · 2018-11-12
> **Updated paper based on your review**
>
> We really appreciate the thoughtful and detailed comments.
> We have corrected the grammatical errors and modified the manuscript based on the suggestions above.
> Specifically, in Section 1, we modified the discerption of MMD; in section 2 we added more information on divergence measures. In section 4 and the supplementary material, we defined constant $c$ as the degeneracy of the U-stats kernel. And for the block estimate, we clarified that the block size is selected to be $\sqrt{n}$ in all our experiments.

---

> > ### Author Response · Authors · 2018-11-15
> > **Updated reference section**
> >
> > Thank you again for your valuable comments.
> >
> > After the first revision, we noticed that the reference section was not updated properly.
> > Thus, we have updated the reference section based on your comments and upload the new version to the system.

---

> > > ### Comment · AnonReviewer3 · 2018-11-19
> > > **Thank you for updating the manuscript; I am happy with the revised version**
> > >
> > > A few additional typos to fix:
> > > -Section 1: 2nd paragraph: 'i.e. higher order' -> 'i.e., higher order',
> > > -Section 2: 1st paragraph: 'larges score' -> 'largest score',
> > > -Section 3.3.: last paragraph: 'see theoretical analysis section' -> 'see Section 4',
> > > -Corollary 3: ',incomplete' -> ', incomplete'.

---

> > > > ### Author Response · Authors · 2018-11-20
> > > > **Fixed typos**
> > > >
> > > > We really appreciate your feedback. We have already fixed typos.

---

### Official Review · AnonReviewer1 · 2018-11-04
**develop post feature selection inference for incomplete mmd**

**Rating:** 5
**Confidence:** 4

**Review:**

The paper propose a method for post feature selection inference in the case where the distribution is non-Gaussian. The paper developed a statistic called incomplete mmd and showed its asymptotic normal property. Then the incomplete mmd can be plugged into post feature selection framework for computing the p-value.

The paper is a nice combination of incomplete mmd and post selection inference technique.
However, the combination is straightforward: the asymptotic Gaussian property of the incomplete mmd is the key.

Furthermore, I think the applications (feature selection and test for GAN objective) is not exciting from the machine learning point of view. A better application which can show-case the obtained p-value is very useful will make the paper more interesting.

---

> ### Author Response · Authors · 2018-11-12
> **Contribution and applicaitons**
>
> >The paper is a nice combination of incomplete mmd and post selection inference technique. However, the combination is straightforward: the asymptotic Gaussian property of the incomplete mmd is the key.
>
> We would like to emphasize that, as far as we know this is the first selective inference method for distribution comparison (or two-sample test). To achieve this, combining MMD and PSI is one of the simple approaches. To deal with the normality constraint in PSI, we also proposed an incomplete U-statistics estimator for MMD and showed basic theoretical properties of the estimator. Although the combination itself is intuitive, since MMD is heavily used in machine learning community in two-sample testing and generative modeling, we believe that this new estimator and application to PSI can shed new light to further improvements of MMD-based algorithms.
>
> >Furthermore, I think the applications (feature selection and test for GAN objective) is not exciting from the machine learning point of view. A better application which can show-case the obtained p-value is very useful will make the paper more interesting.
>
> Feature selection (or variable selection) is one of important machine learning problems including sparse learning. For instance, Lasso is a widely known feature selection method and there exist a large number of related papers in top machine learning venues. Even among the ICLR submissions this year, we found an another interesting feature selection paper:
> KnockoffGAN: Generating Knockoffs for Feature Selection using Generative Adversarial Networks
> https://openreview.net/forum?id=ByeZ5jC5YQ
>
> Moreover, feature selection has a number of important applications in biology and healthcare. In these filed, estimating p-value is extremely important. As for comparing generative models, analyzing GAN is one of hot topics in machine learning in particular for deep learning communities; although the result that we get (all null hypotheses rejected) is not the most surprising, we believe that the problem itself is important and further study needed.

---

### Official Review · AnonReviewer2 · 2018-11-06
**Could be an important contribution but not clear from the paper**

**Rating:** 6
**Confidence:** 4

**Review:**

The paper proposes a new post selection inference for MMD statistics, i.e., identity the p-values for the dimensions of vector. I believe this is an important problem that has not been addressed in previous literature. The work provides an extension to the original post selection inference work for lasso (Lee et al. 2016).

However, I wish the paper could have explained the main idea clearly. Right now it is hard to me to judge whether or not the proposed estimator is correct (the only place that seems to support this is Fig. 4(a). Where the p-value seems to be clear to a uniform distribution.

For instance, the proposed PSI estimator, we will need to estimate the covariance matrix. This was explained in Section 3.3, and it was said that the algorithm in Fan et al. (2013) was used for the explanation. However, I think more detailed discussions and explanation should be provided here. In order to obtain correct p-value estimate, I believe getting accurate covariance matrix estimate is crucial. How large the sample size is needed, in order for us to get an accurate enough covariance matrix, to perform sub-sequent post selection inference? More discussions are needed here.

---

> ### Author Response · Authors · 2018-11-12
> **About covariance matrix estimation**
>
> Thank you for your valuable comments.
>
> >However, I wish the paper could have explained the main idea clearly. Right now it is hard to me to judge whether or not the proposed estimator is correct (the only place that seems to support this is Fig. 4(a). Where the p-value seems to be clear to a uniform distribution.
>
> The main idea is to use an estimator of MMD with normal response to perform testing and selective inference. The proposed incomplete estimator randomly subsamples the U-stats kernels of MMD, and asymptotic normality follows from properties of incomplete U-statistics. In finite sample cases, the correctness of this estimator is supported by the uniform p-value distribution in Figure 4(a), and also the fact that the false positive rate is successfully controlled at the desired \alpha in both synthetic and real datasets.
>
> >For instance, the proposed PSI estimator, we will need to estimate the covariance matrix. This was explained in Section 3.3, and it was said that the algorithm in Fan et al. (2013) was used for the explanation. However, I think more detailed discussions and explanation should be provided here. In order to obtain correct p-value estimate, I believe getting accurate covariance matrix estimate is crucial. How large the sample size is needed, in order for us to get an accurate enough covariance matrix, to perform sub-sequent post selection inference?
>
> For hypothesis testing, the consistency of the covariance matrix estimator is necessary to get accurate covariance matrix estimates. In our PSI setup, the consistency of the covariance matrix estimator which corresponds to the result of Theorem 2 and its corollaries is derived under standard regularity conditions.
>
> In practice, for the small number of samples, the estimation of covariance matrix is extremely hard. As shown in the Figure 6 of supplementary material, we experimentally found that the estimation error of the incomplete MMD covariance is smaller than that of the block MMD covariance. Moreover, for the incomplete U-statistics MMD, we have $\ell=rn$ samples to estimate the covariance matrix, while we have only $n/B$ samples for the block MMD. That is, the number of samples that can be used for covariance estimation of the incomplete U-statistics estimator is $rB$ times larger than that of the block estimator. In the setup of Figure 6, we have $rB = 100$. For example, for $n = 1000$, we can use $5000$ samples for the incomplete U-statistics, while we have only $50$ samples for the block estimator. Thus, for the block MMD, this is problematic for high-dimensional case (e.g., d = 100), and the estimated covariance matrix is not full-rank. To alleviate this issue, we employ the POET algorithm for the block estimator.
>
> Moreover, we run the PSI algorithms for the white wine dataset by changing the number of samples as 200, 400, …, 4000. As can be seen, both PSI algorithms can control the FPR with 2000 samples. Moreover, it is clear that the incomplete U-statistics estimator can get better performance than the block estimator in both TPR and FPR.  We included this experimental result in the revised supplementary material (see Figure 8).

---

### Meta-Review · Area_Chair1 · 2018-12-16
**Area chair recommendation**

**Confidence:** 5
**Recommendation:** Accept (Poster)

**Metareview:**

The submission evaluates maximum mean discrepancy estimators for post selection inference.
It combines two contributions: (i) it proposes an incomplete u-statistic estimator for MMD, (ii) it evaluates this and existing estimators in a post selection inference setting.

The method extends the post selection inference approach of (Lee et al. 2016) to the current u-statistic approach for MMD.  The top-k selection problem is phrased as a linear constraint reducing it to the problem of Lee et al.  The approach is illustrated on toy examples and a GAN application.

The main criticism of the problem is the novelty of the paper.  R1 feels that it is largely just the combination of two known approaches (although it appears that the incomplete estimator is key), while R3 was significantly more impressed.  Both are senior experts in the topic.

On the balance, the reviewers were more positive than negative.  R2 felt that the authors comments helped to address their concerns, while R3 gave detailed arguments in favor of the submission and championed the paper.  The paper provides an additional interesting framework for evaluation of estimators, and considers their application in a broader context of post-selection inference.